# Radioactivity Monitoring at North Aegean Sea Integrating In-Situ Sensor in an Ocean Observing Platform

**Christos Tsabaris \***[iD]**, Effrossyni G. Androulakaki, Dionysios Ballas, Stylianos Alexakis, Leonidas Perivoliotis and Athanasia Iona**

Hellenic Centre for Marine Research, Institute of Oceanography, P.O. BOX 712, Gr-19013 Anavyssos, Greece; frosso.androulakaki@hcmr.gr (E.G.A.); dballas@hcmr.gr (D.B.); salexakis@hcmr.gr (S.A.); lperiv@hcmr.gr (L.P.); sissy@hcmr.gr (A.I.)

\* Correspondence: tsabaris@hcmr.gr; Tel.: +30-22910-76410; Fax: +30-22910-76323

**Abstract:** The integration of the radioactivity spectrometer KATERINA II in a fixed station (buoy) of the POSEIDON network at the North Aegean Sea within the framework of MARRE Project is presented. The acquisition period lasted from 20 November 2019 till 22 February 2020. An intense increment of the activity concentration of radon progenies (up to an order of magnitude) was recorded during rainfall. More specifically, the $^{214}$Bi activity concentration varied from 0.09 to 0.53 Bq L$^{-1}$ without rainfall and the $^{214}$Pb activity concentration varied from 0.14 to 0.81 Bq L$^{-1}$. The $^{214}$Bi activity concentration during rainfall ranged from 0.4 to 5.4 Bq L$^{-1}$ and of $^{214}$Pb from 0.3 to 5.3 Bq L$^{-1}$. The minimum detectable activity of the KATERINA II detection system for measuring low level activities of $^{137}$Cs is optimized applying background subtraction and the full spectrum analysis technique.

**Keywords:** radioactivity; underwater sensors; ocean buoys; rainfall; North Aegean Sea





## 1. Introduction

Radioactivity monitoring in the marine environment is widely used for monitoring natural and artificial contributions, which are attributed to radiological incidences occurring in nuclear and non-nuclear industries close to the coastline. An example of possible artificial radioactivity spread across Europe is the recent fire that took place in the Chernobyl exclusion zone.

The development of systems for the detection of radioactive contamination has been expanded the last 20 years by applying innovative cost-effective tools (hardware and measurement method) in order to efficiently support the responsible authorities and the scientific radioecological communities. Different in situ methodologies and tools regarding radioactivity monitoring have been tested in many marine areas of the world the last years [1–11], depending on the selected area, topography and contamination level. The continuous monitoring is the only recommended method in cases that the activity concentration of radionuclides is above the minimum detectable activity of the in-situ detection system.

Assuming that the background gamma-ray radiation level (which depends mostly on the water salinity) is constant at a specific area at the open sea, the fluctuations of the gross counting rate in seawater are mainly caused by rainfall due to the atmospheric wash out of the various natural radionuclides (in the absence of a radiological incidence). Radon as a natural radioactive inert gas and non-chemically reactive element is studied extensively in the terrestrial and marine environment concerning NORM and TENORM studies [12]. One of its isotopes, $^{222}$Rn, although it is not a gamma-emitter, can be detected via gamma-ray spectrometry from its progenies ($^{214}$Bi and $^{214}$Pb). These two radionuclides have been extensively utilized as radiotracers in oceanography applications [13,14],

such as water masses mixing, radioprotection purposes in non-nuclear industries (fertilizers, oil exploration, oil and construction industries), pockmarks [15], mud volcanoes [16], and submarine groundwater discharge identification [17,18]. Moreover, the monitoring of natural and artificial radioactivity variations during rainfall at the sea can provide important information in critical hazards (e.g., fires, floods). The radionuclides $^{214}$Bi and $^{214}$Pb have been used as radiotracers also in different scientific fields. Data regarding the utilization of radiotracers for meteorological studies (e.g., correlation rainfall events with wind speed direction) is scarce [6,19,20], especially in the marine environment, although it provides ideal conditions for such studies, eliminating factors adding complexity (e.g., wind, temperature variations). In these studies, strong correlations of rainfall events with radon progenies activity concentrations have been observed [6,20], depending on the passage and origin of the rainfall (if it comes from the terrestrial or marine environment). Therefore, monitoring data of radionuclide concentrations at the sea is requested as baseline information, so any possible environmental variations could be identified.

In this study, the correlation analysis between radioactivity data and rainfall parameters from neighboring stations was performed. Moreover, the phenomenon of surface salinity variations induced by rainfall dilution [21,22] was observed via radioactivity data using $^{40}$K as a tracer. The main goal of this work was to study the atmospheric wash out due to rainfalls, regarding natural (e.g., radon progenies) and artificial radioactivity (e.g., $^{137}$Cs), as well as to improve the minimum detectable activity (MDA) of the detection system in order to detect low concentrations of the artificial $^{137}$Cs. For this purpose, a new methodology is developed, which relies on spectra subtraction to minimize the measurement background prior to the analysis and utilization of the Full Spectrum Analysis (FSA) technique for the $^{137}$Cs concentration values estimation. To this end, monitoring data in the Athos region at the North Aegean Sea were obtained, for studying radioactivity fluctuations during rainfall events, in the period from 20 November 2019 to 22 February 2020.

In the next paragraph, the floating measuring system (buoy), together with the system upgrade are described. The methodology of data analysis, the variability of the activity concentration of natural radionuclides due to rainfall events and their interpretation are also described. Moreover, the results of the developed methodology for optimizing the MDA of $^{137}$Cs (artificial radionuclide) for the KATERINA II system [6] are given for long (24 h) and short-term (3 h) periods of acquisition.

## 2. Materials and Methods

### 2.1. The Fixed Mooring System (Athos Station)

The POSEIDON system is an observatory research infrastructure that aims to establish a sustainable marine observing network in the Eastern Mediterranean and to provide quality and validated forecasts for the marine environment. The POSEIDON observing component consists of a network of moored buoys, two underwater gliders, a ferrybox system, a cabled observatory, a fleet of Argo floats, an HF radar system, and a repeated cruises-based ecosystem monitoring program. The system has been in operation for more than 20 years and was developed in accordance to the policy frameworks suggested by IOC/GOOS, EuroGOOS, MonGOOS and GEO. Since 2015, it has participated in the Copernicus Marine Environment Monitoring System (CMEMS—marine.copernicus.eu), providing the near real time in situ data and the wave forecasts for the Mediterranean Sea [23,24]. The POSEIDON mooring network consists of six fixed stations deployed in Aegean and Ionian Seas (Pylos station in SE Ionian, Athos station in North Aegean Sea, Mykonos station in Central Aegean, Saronikos station in the gulf that surrounds the Athens metropolitan area and two stations in South Aegean—Cretan Sea). The Athos station has operated since 2000, carrying recording sensors for the surface marine and atmospheric conditions. In the neighborhood of this station, the National Observatory of Athens owns and maintains two meteorological stations in Vatopedi area and Lemnos Island. The marine and terrestrial stations are shown in Figure 1.

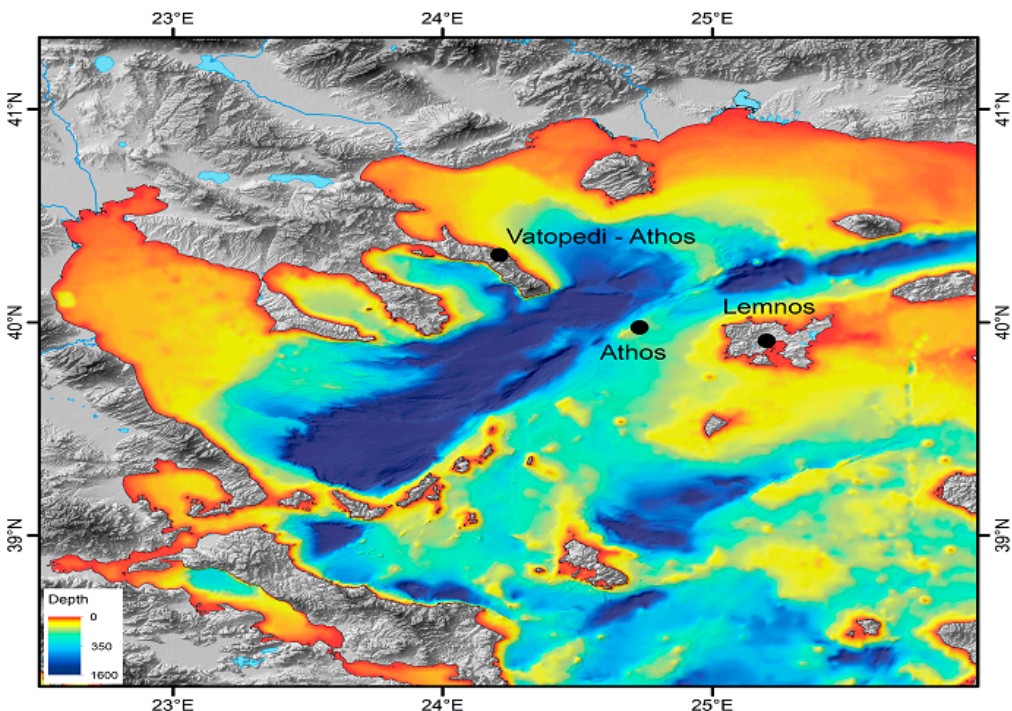

**Figure 1.** The area of study.

A typical fixed mooring consists of steel wire or synthetic rope, which holds the integrated sensors, and is associated with one or more buoys to provide sufficient buoyancy to keep the mooring upright. Each fixed platform has different kinds of sensors at different depths, providing atmospheric and marine data in the water column. The data recorded by the moorings are transmitted in real-time using GPRS and/or satellite telecommunication links. The long-term recordings collected by the fixed stations can provide evidence on the ocean state and ocean variability, allow a better understanding on the several processes that take place in the marine environment, support the calibration and verification of the marine and weather forecasting systems and enhance the maritime safety, and the efficient planning of marine infrastructures. The mooring network is maintained regularly (twice a year) through dedicated cruises realized by the HCMR's research vessel.

Although the power system of the station provides continuous energy for all sensors, an independent battery pack was installed for the period of measurements using the radioactivity sensor that was operated in stand-alone mode. Using the software of the installed radioactivity sensor, data are automatically saved according to the time lag of the system. A schematic view of the ocean observing system at Athos with the installed radioactivity sensor and the power supply unit are shown in Figure 2.

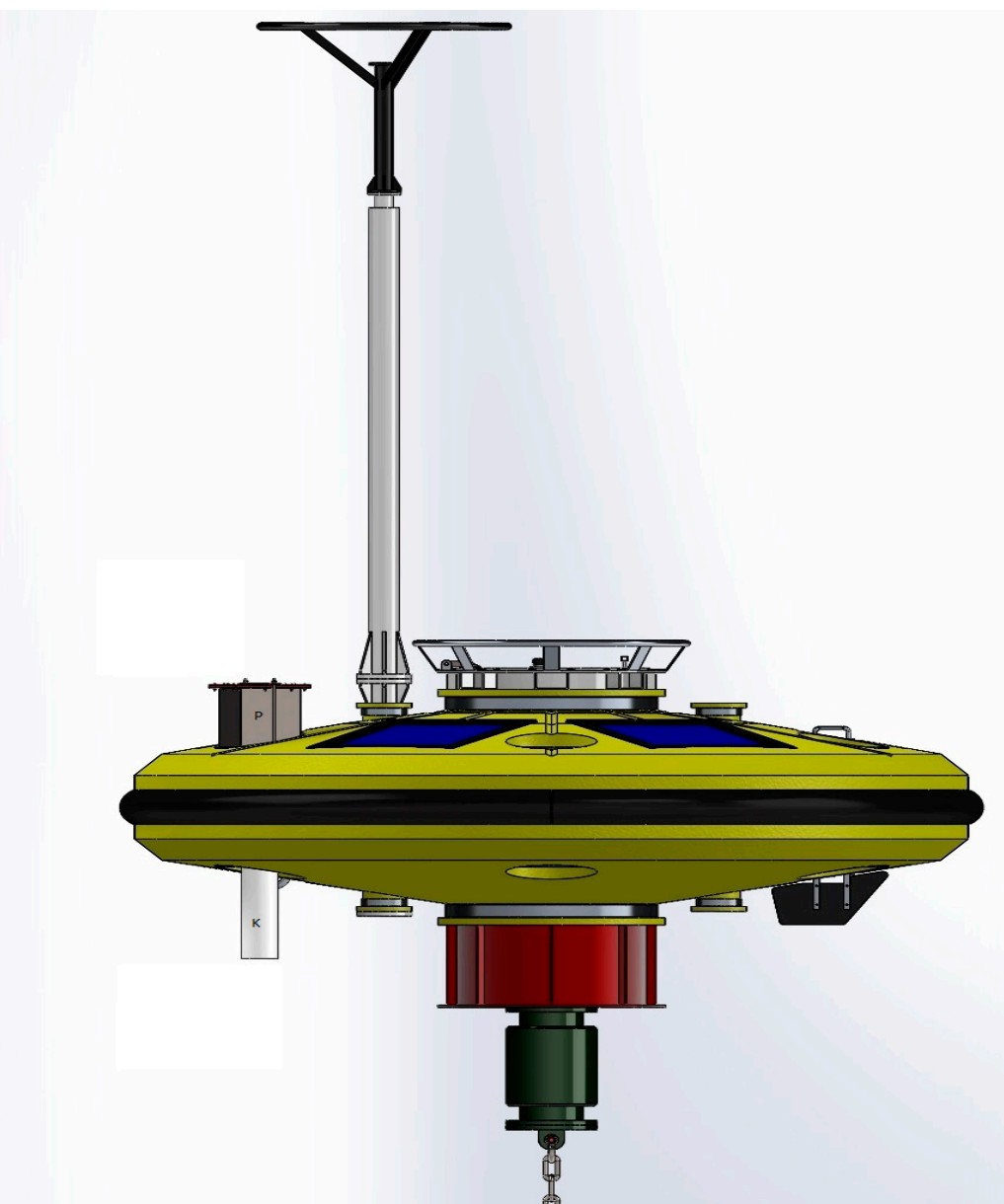

**Figure 2.** The POSEIDON fixed station at Athos (North Aegean Sea). The sensor is represented with "K" and the power supply unit with "P".

## 2.2. Acquisition and Data Analysis

The radioactivity sensor KATERINA II is integrated in the floating observing system of the POSEIDON network, and the crystal was located at a depth of 70 cm below the sea surface. KATERINA II is a compact and autonomous underwater gamma-ray spectrometer capable of monitoring natural and artificial radionuclides. A scintillation crystal of NaI(Tl) (7.62 × 7.62 cm) is used providing spectra of gamma-rays in a wide range of energy (50–2800 keV) based on a multi-channel analyzer with varied spectroscopy. The system resolution is ~6.5% for $^{137}$Cs (662 keV) detection. The dead time is less than 0.5% when the system operates in seawater. All parts of the system are tightly assembled inside a cylindrical housing made of acetal, allowing maximum operation depth at around 400 m. The KATERINA II system is designed to be integrated into measuring fixed ocean platforms. The system also has the specifications to operate in stand-alone mode as well as in sequential buffering acquisition mode.

The electronics of the shaping amplifier provide stable voltage output, compensating voltage drifts (which are the typical drawback of scintillation-based gamma-ray spectrometers). The intense peak of [40]K at 1461 keV was always observed at the middle of the spectrum. Potential voltage drifts (especially in the air) are compensated via gain-stabilization procedure using the two present peaks at the energies of 50 and 1461 keV [25,26].

An important upgrade of the KATERINA II system with respect to previous versions is the standalone operation of the system providing time series of data without connection to a computer. Appropriate software was developed to perform measurements in the pre-defined time lag for a specific period of time, saving all data in a volatile memory [6]. The measured data were automatically saved mainly every 3 h of time lag in the aforementioned memory as raw measurements (as it was programmed before the acquisition start). Thus, after the end of the acquisition period, the gamma-ray spectra are automatically stored inside the memory and are retrieved every 2 months (using computer connection). The data pass first through a quality control by checking potential spikes and then a second quality control that identifies abnormal trends (with or without rainfalls).

Before the installation of the detection system at the ocean observing system (buoy), the KATERINA II system was calibrated (for energy, energy resolution and efficiency) in the energy range from energy threshold to 2800 keV for quantitative results as described elsewhere [6,25,27]. The first step in the analysis is the energy calibration as given from existing methodology at the laboratory [27,28]. The data analysis was performed by subtracting the background spectrum (the measured spectrum before rain fall) from the measured spectrum during rainfall. Several tests were made by comparing measured spectra acquired during dry periods, in order to specify a global background spectrum. This was possible due to the constant counting rate in dry periods and the stability of the measured spectra (since spectra subtraction is not applicable if voltage drifts take place during measurements). The spectrum analysis was performed using the software package SPECTRW [29].

The activity concentration of radon progenies ([214]Bi, [214]Pb) was calculated in rainfall events, using experimental and theoretical techniques. The SPECTRW software [29] was utilized in the spectra analysis. The activity concentration of radon progenies was deduced taking into account the detection efficiency of the system, the detected counting rate, the acquisition time, and the emission probability of the detected gamma-rays [27,28]. The statistical uncertainty of the simulation runs for the detection efficiency of the system was kept below 4% in the energy regions of the photopeaks, in all cases. Thus, the total uncertainty budget is calculated using the typical propagation law of relative uncertainties taking into account the uncertainties of the system efficiency (4%) and the statistical counting rate uncertainty in the analyzed photopeaks (4–39%). The analysis is performed assuming homogenized enrichment of radon progenies in the seawater and rapid dilution process in the time lag window of the detection system (3 h).

### 2.3. Full Spectrum Analysis (FSA) Technique

The typical analysis technique is the photopeak analysis. However, there are some critical drawbacks due to the overlapping energy peaks and the introduction of systematic uncertainty related to background subtraction issues. The most important drawback is the inability of the method to identify unexpected radionuclides [30] that may contribute to the measured number of events in the selected energy window. Consequently, the Full Spectrum Analysis (FSA) technique was also implemented in the analysis of those spectra obtained during rainfall events. The FSA technique is based on the reproduction of standard spectra for all radionuclides of interest. Standard spectra are obtained either experimentally by performing calibration measurements [30,31], or theoretically by performing Monte Carlo (MC) simulations [32,33], including the geometry and detailed characteristics of the measurement. In this work, the standard spectra were derived using the MCNP-CP code [34]. The first step for the spectra analysis using the FSA technique, in order to derive the activity concentrations, was the subtraction of the background radiation (normalized

spectrum during periods without rainfall events) from the spectra during rainfall to exclude the $^{40}$K and cosmic radiation contributions. Then, the $^{214}$Bi and $^{214}$Pb standard spectra were normalized to the experimental conditions (electronic setup, acquisition time) and were fitted to the experimental spectrum to derive the activity concentrations. Details of the FSA technique and the convergence between experimental and theoretical spectra are given in the literature [35]. Nevertheless, it is important to mention that in the simulations the radionuclides are considered to be homogeneously distributed in a large (with a radius of 50 cm to 200 cm depending on the detected gamma-ray energy) spherical volume surrounding the detector [35,36].

## 3. Results

### *3.1. Quality Control of γ-Ray Spectra*

In the period from 20 November 2019 to 22 February 2020, gamma-ray spectra were carried out with the subsea detection system KATERINA II at the Athos station of POSEIDON network. The KATERINA II sensor provided a measurement (γ-ray spectrum) every 3 h from 20 November 2019 till 25 January 2020 and every 6 h from 12 February 2020 till 22 February 2020. Prior to the analysis, a quality control was performed to evaluate the collected data and search for possible malfunctions leading to poor quality data.

In this work, the system exhibited an excellent performance, leading to a total of 428 acquired spectra. A satisfactory observation was that the detection system exhibited excellent stability (less than five channels of voltage drifts), assisting effective spectrum analysis. Typical gamma-ray spectra, that were acquired in periods with rainfall (e.g., 12 December 2019) and without rainfall (defined as background spectra), are depicted in Figure 3. More specifically, the spectra obtained during and after rainfall on 12 December 2019 (3 h measurements) are depicted together with the background spectrum (3 h measurement and without rainfall).

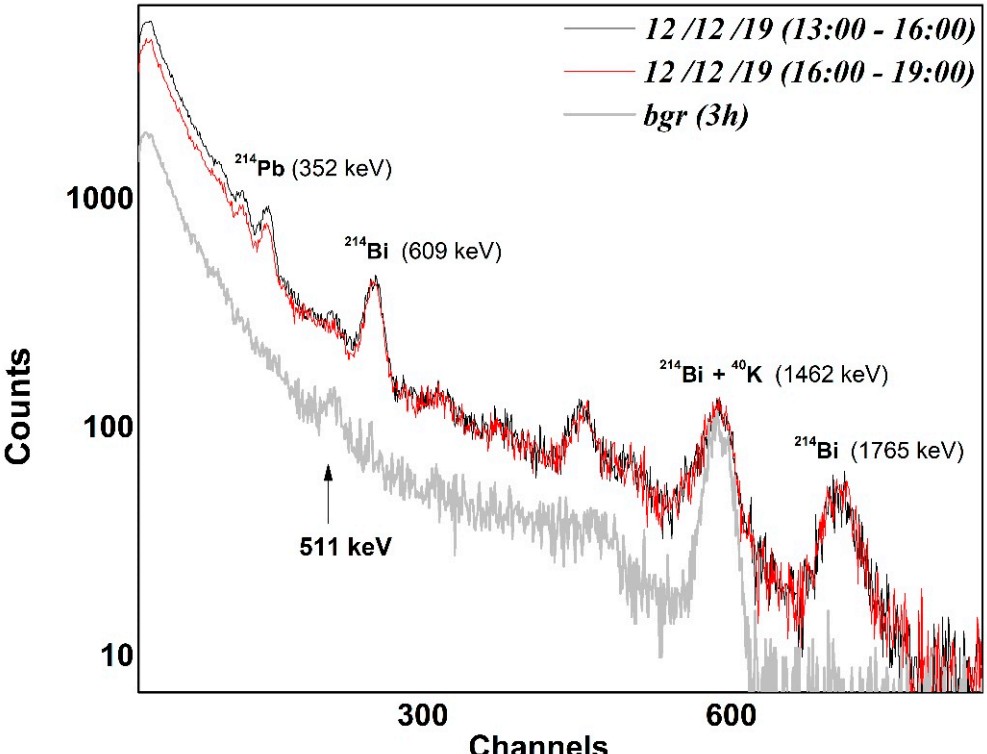

**Figure 3.** The background spectrum without rainfall and two spectra with rainfall. The acquisition period was 3 h.

The background spectrum was acquired in a period without rainfall close to the afore-mentioned date (12 December 2019). The data are depicted in arbitrary units of counts per second along with energy (channels) since there was no need for energy calibration exercises due to the stable output signals. The main characteristic of the measured spectra without rainfall is that the gross counting rate of the system varies in the limit of the statistical uncertainty. This value is considered as a background value of the gross gamma-ray intensity of the KATERINA II sensor operated at the specific site of the sea. Each variation that exceeded the statistical uncertainty was considered as a rainfall event, if the salinity (detected from the $^{40}$K peak) remained constant.

However, after a rainfall event, several photo-peaks attributed to radon progenies were identified in the spectrum (Figure 3). More specifically, photo-peaks of the gamma-ray emitters of radon ($^{214}$Bi and $^{214}$Pb) are depicted as raw data along with energy (channels). The $^{214}$Pb radon progeny is observed at the gamma-ray energies at 241, 295 and 352 keV, while the $^{214}$Bi radon progeny is identified mainly at the energies of 609 and 1764 keV (gamma rays of high intensity). The aforementioned gamma-rays are clearly seen in the spectrum after rainfall. At 12 December 2019 (13:00–16:00), the spectra provide similar activity concentration (within uncertainties) of the two radon progenies, while at 12 December 2019 (16:00–19:00), the counting rate of $^{214}$Bi between the two spectra is slightly varied. At the same spectrum, the counting rate of $^{214}$Pb is reduced during the second period (16:00–19:00). This reduction of the $^{214}$Pb activity concentration between the two continuous periods of acquisition may be attributed to its decay, since rainfall intensity is also reduced.

In all spectra, an intense contribution at 511 keV is also observed from the detection system. The observed peak in this energy is attributed to the mechanism of the pair production, which may occur when highly energetic γ-rays travel along the seawater (close to the detector) or from the atmosphere (since the detector is placed relatively close to the seawater surface). A pair production event is observed through the accompanied appearance in the spectrum of the annihilation peak (511 keV). Such events are not present when the detector is deployed in high depths (above 10 m), due to the screening of the energetic γ-rays in the seawater column [6].

### 3.2. Gross Gamma-Ray Intensity Analysis

The observations of rainfall events (rain in mm normalized to the radioactivity sensor lag time) as given from the POSEIDON model and the two meteorological stations in Lemnos and Vatopedi [37], are depicted in Figure 4a,b, along with the gross counting rate recorded using the KATERINA II sensor. The data are depicted for the whole monitoring period and for the period from 20 November 2019 to 20 December 2019 (see Figure 4). In most cases, a high increase of the gross counting rate is observed during rainfall events.

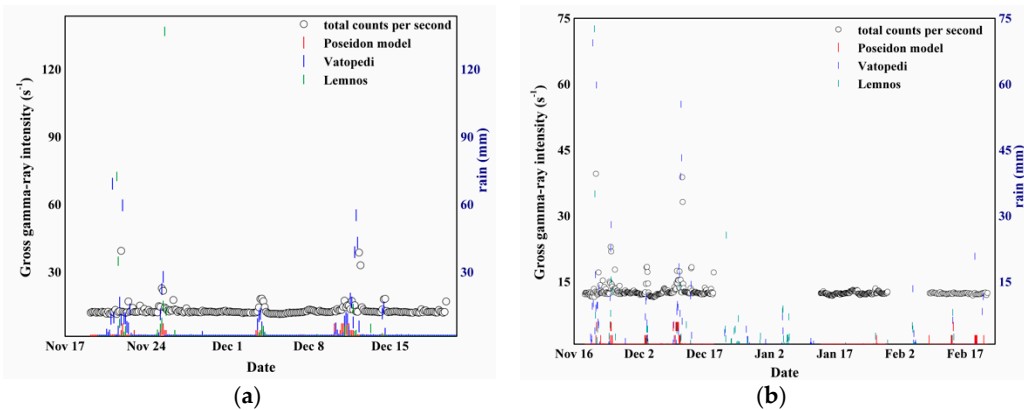

**Figure 4.** The counting rate (as given from the KATERINA II system) and the rain intensity from Poseidon model and Lemnos and Vatopedi stations for two periods: 20/11/2019–22/02/2020 (**b**) and from 20/11/2019–20/12/2019 (**a**).

The gross counting rate of the recorded events ranges between 11.6 events per second and 13.0 events per second in dry periods (without rain) and increases up to four times during and after rainfall events (the maximum value reaches 40 events per second). Earlier measurements [5] in the same area from May 2000 to December 2002 exhibited similar values with the total recorded event rate ranging from 9 to 12 events per second in dry periods.

During the observation period, 15 events with high gross counting rate were observed with the gross counting rate ranging from 17 events per second to 40 events per second detected during the period 20 November 2019 to 20 December 2019. In the monitoring period from 20 November 2019 to 22 February 2020, 79 events of rainfall took place, providing a gross counting rate larger than 13 events per second or cps ($s^{-1}$).

The rainfall events that provide gross counting rates above the background value are correlated with rain signals from the terrestrial stations [37], although they are distant from the marine station. The rainfall events are also identified by the POSEIDON forecasting model [23,24]. Very few measurements exhibited high counting rate (cps~$17s^{-1}$) in the buoy station, while the terrestrial neighboring meteorological stations did not record any signal (possibly a local low intensity rainfall took place in the area of the buoy without affecting the Vatopedi and Lemnos stations). An accurate correlation of the gross counting rate with the rainfall events is not applicable due to the lack of experimental rain intensity data from the same position of the KATERINA II sensor (since the rainfall intensity data are taken from POSEIDON model and from neighboring stations in the terrestrial environment).

An interesting result from the data analysis is that after 20 December 2019, the detected events per second (cps), as given from the KATERINA II sensor, are reduced during rainfalls. This could be associated with a change of the trend of the wind direction (preferably south winds were present). Radon as an inert gas is emanated from the Earth's crust and then transported by wind and air advection to the observing station. Consequently, if the wind direction is not originated from the terrestrial part, the radon concentration is drastically reduced, since radon progenies' concentrations in the seawater are negligible.

### 3.3. Radioactivity Analysis

The results of the activity concentration of radon progenies (3h spectra) together with the meteorological data of neighboring stations are given in Table 1. Only values above 0.4 Bq $L^{-1}$, for which the statistical uncertainty is tolerable, are presented using the commercial software package (SPECTRW). The FSA method was applied in cases where the activity concentration was above 1 Bq $L^{-1}$, since the statistics below this value in the subtracted spectra were insufficient.

In most cases, the activity concentration of $^{214}$Bi was in good agreement (within uncertainties) with the activity concentration of $^{214}$Pb. The activity concentration of $^{214}$Bi on rainfall events varies from 0.4 to 5.4 Bq $L^{-1}$, while for $^{214}$Pb from 0.3 to 5.3 Bq $L^{-1}$. The activity concentration of radon progenies ($^{214}$Bi and $^{214}$Pb) is higher during East winds at North Aegean Sea, in which rainfall is transferred from the terrestrial part of North Greece and not from the marine areas. The same behavior was observed also in the past at the same station [5].

**Table 1.** The activity concentration of radon progenies ($^{214}$Bi and $^{214}$Pb) during rainfalls analysed with the SPECTRW software and the FSA method. The time-normalized meteorological data of two neighboring stations (Vatopedi and Lemnos) are also included.

| | | SPECTRW Analysis | | FSA Analysis | | Vatopedi Station | | Lemnos Station | |
|---|---|---|---|---|---|---|---|---|---|
| Date | Time | $^{214}$Bi (Bq l$^{-1}$) | $^{214}$Pb (Bq l$^{-1}$) | $^{214}$Bi (Bq l$^{-1}$) | $^{214}$Pb (Bq l$^{-1}$) | Rain (mm) | Wind Direction | Rain (mm) | Wind Direction |
| 22/11/2019 | 1:00 | 5.2 (4) | 5.2 (4) | 5.2 | 4.8 | 9.2 | ESE | 0.8 | NE |
| 25/11/2019 | 13:00 | 2.6 (4) | 2.6 (4) | 2.9 | 2.2 | 22.8 | ESE | 8 | ENE |
| 25/11/2019 | 16:00 | 2.5 (4) | 2.5 (4) | 2.7 | 2.3 | 28.2 | E/ESE | 15.6 | ENE |
| 26/11/2019 | 13:00 | 1.4 (4) | 1.2 (5) | 1.7 | 1.4 | 0 | NNE/ESE | 0.2 | SW/NW |
| 27/11/2020 | 16:00 | 0.7 (17) | 0.8 (19) | | | 0 | - | 0 | SE |
| 4/12/2019 | 1:00 | 0.8 (12) | 0.9 (18) | | | 11.8 | ESE | 2.2 | NE |
| 4/12/2019 | 4:00 | 1.4 (10) | 1.3 (21) | 1.7 | 1.4 | 5.8 | SE | 4.4 | NNE |
| 4/12/2019 | 7:00 | 1.2 (9) | 1.2 (14) | 1.4 | 1.3 | 2.4 | ESE | 4.4 | NE |
| 4/12/2019 | 10:00 | 0.8 (13) | 0.9 (19) | | | 1.2 | ESE | 1.4 | NNE |
| 4/12/2019 | 13:00 | 0.7 (12) | 0.8 (16) | | | 0.2 | ESE | 0.4 | NNE |
| 8/12/2019 | 4:00 | 0.5 (16) | 0.5 (24) | | | 0 | WNW | 0.4 | WNW |
| 9/12/2019 | 4:00 | 0.5 (18) | 0.8 (16) | | | 0 | WNW | 0.2 | NW |
| 10/12/2019 | 22:00 | 0.5 (16) | 0.5 (27) | | | 0 | ESE | 0 | NE |
| 11/12/2019 | 1:00 | 0.7 (13) | 0.3 (38) | | | 0 | ESE | 0 | NE |
| 11/12/2019 | 7:00 | 0.6 (29) | 0.4 (32) | | | 8.4 | ESE | 4.2 | NE |
| 11/12/2019 | 10:00 | 1.0 (13) | 1.1 (15) | 1.1 | 1.2 | 10 | SE | 5.2 | NE |
| 11/12/2019 | 13:00 | 0.6 (15) | 0.3 (39) | | | 9.4 | SE | 0.2 | NE |
| 11/12/2019 | 16:00 | 0.7 (16) | 0.4 (35) | | | 3.8 | SE | 0 | NE |
| 11/12/2019 | 19:00 | 0.7 (14) | 0.7 (19) | | | 18.6 | SE | 0 | NE |
| 11/12/2019 | 22:00 | 0.7 (12) | 0.6 (21) | | | 16.8 | ESE | 0.8 | NE |
| 12/12/2019 | 1:00 | 1.1 (8) | 0.6 (30) | | | 8 | ESE/SE | 14.2 | NE |
| 12/12/2019 | 4:00 | 0.9 (10) | 0.9 (16) | | | 39.2 | SE | 2.2 | E/NE |
| 12/12/2019 | 10:00 | 0.7 (12) | 0.6 (28) | | | 43.4 | SSE | 0 | NE/E |
| 12/12/2019 | 13:00 | 5.2 (4) | 5.3 (4) | 5.2 | 4.8 | 6.2 | - | 0 | E/ENE |
| 12/12/2019 | 16:00 | 5.4 (4) | 5.2 (4) | 5.2 | 4.3 | 0 | NW | 0 | NE |
| 12/12/2019 | 19:00 | 1 (10) | 1.1 (13) | | | 0.2 | NW | 0 | NE |
| 12/12/2019 | 22:00 | 0.6 (16) | 0.4 (30) | | | 0.2 | S | 0 | NNW |
| 13/12/2019 | 1:00 | 0.5 (18) | 0.7 (22) | | | 0 | S | 0.2 | NNW |
| 14/12/2019 | 16:00 | 0.7 (14) | 0.8 (19) | | | 14.6 | ESE/SE | 0 | ESE/SE |
| 14/12/2019 | 19:00 | 1.4 (8) | 1.9 (10) | 1.7 | 1.9 | 3 | NNW | 0.2 | ENE |
| 14/12/2019 | 22:00 | 1.4 (8) | 1.8 (10) | 1.9 | 1.8 | 0 | NNW | 0.4 | WNW |
| 15/12/2019 | 1:00 | 0.7 (15) | 0.5 (28) | | | 0 | SSW | 0 | WNW |
| 20/12/2019 | 1:00 | 1 (20) | 1.2 (11) | | | 0 | - | 0 | - |
| 27/1/2020 | 9:30 | 0.6 (14) | 0.8 (16) | | | 0 | WSW | 0 | SSW |
| 27/1/2020 | 12:30 | 0.6 (13) | 0.7 (17) | | | 0 | WSW | 2.8 | SSE |
| 27/1/2020 | 15:30 | 0.5 (14) | 0.5 (23) | | | 0 | SSW | 0.4 | SW |
| 27/1/2020 | 18:30 | 0.5 (15) | 0.5 (24) | | | 0 | SSW | 0 | SW |
| 28/1/2020 | 3:30 | 0.5 (16) | 0.6 (21) | | | 0 | WNW | 0 | - |
| 30/1/2020 | 0:30 | 0.5 (17) | 0.5 (23) | | | 0 | WNW | 0 | WSW |
| 30/1/2020 | 3:30 | 0.4 (23) | 0.4 (23) | | | 0 | WNW | 0 | WNW |
| 30/1/2020 | 6:30 | 0.4 (20) | 0.5 (23) | | | 0 | NW | 0.2 | WNW |
| 30/1/2020 | 9:30 | 0.5 (15) | 0.5 (20) | | | 0 | NNW | 1.2 | NE |

The activity concentrations of radon progenies using the SPECTRW software package and those deduced using the FSA method, were found in good agreement (within uncertainties). The reproduced theoretical spectrum (using the FSA technique) together with the experimental spectrum (after the appropriate background subtraction) of a rainfall event (between 21 to 22 November 2019, summation of nine acquired spectra normalized to 3 h), are shown in Figure 5. The experimental spectrum is fitted using standard spectra of 0.3 Bq L$^{-1}$ activity concentrations for $^{214}$Bi and $^{214}$Pb. The agreement between the reproduced with the experimental spectrum is satisfactory along with the detection energy taking into account the statistical fluctuations. This agreement implies that the theoretical model describes satisfactorily the measurements as given by the KATERINA II system taking into account all the mechanisms of interaction of gamma rays with water and detector materials. The observed fluctuations may be attributed to small differences between the experimental electronic setup (energy calibration, energy resolution calibration) and the corresponding values introduced in the theoretical calculations. The analysis of other radionuclides, such as $^{40}$K and $^{137}$Cs, is also performed. As concerns the $^{40}$K results, the data exhibit stable activity concentration values during dry periods representing the salinity values of the seawater.

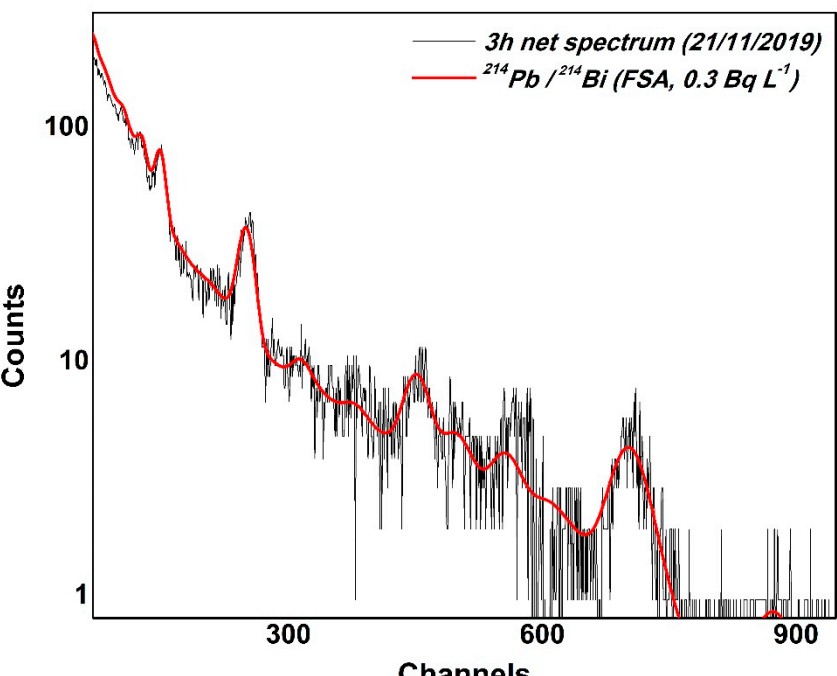

**Figure 5.** The experimental and reproduced spectra (after subtraction of the background spectrum) of the radon progenies ($^{214}$Pb, $^{214}$Bi). The activity concentration for both progenies is 0.3 Bq L$^{-1}$, and the acquisition time is 3 h.

However, the activity concentration of $^{40}$K is drastically decreased during and after strong precipitation due to the rapid mixing processes of the rainwater with the seawater. A typical example is given in Figure 6, where the influence of precipitation is clearly observed from the KATERINA II detection system by the decrease of the measured $^{40}$K activity concentration (almost 30% compared with the dry periods) during the precipitation event on 22 November 2019. The same behavior was observed during the rainfall event on 12 December 2019. This precipitation had an amount of 9.2 mm, and the rainwater was enriched with radon progenies ($^{214}$Bi) concentration of (5.2 ± 0.2) Bq L$^{-1}$. The calculated salinity values using the $^{40}$K activity concentrations ranged from 30 to 39 psu, while similar values ranging from 33 to 37 psu were also obtained from the buoy CTD measurements (in 1m depth) in the same station. This was expected as the $^{40}$K concentration is very well correlated with surface salinity values in the seawater [13]. The salinity data (both the

values and the observed variations) agree with salinity values obtained in the past in the same station [38]. The decrease of the $^{40}$K activity concentration could be attributed to the dilution of seawater through mixing with different water masses either due to the large amount of rainwater or a passage of a seawater current mass (e.g., Black Sea Water), which has been observed in the past [38].

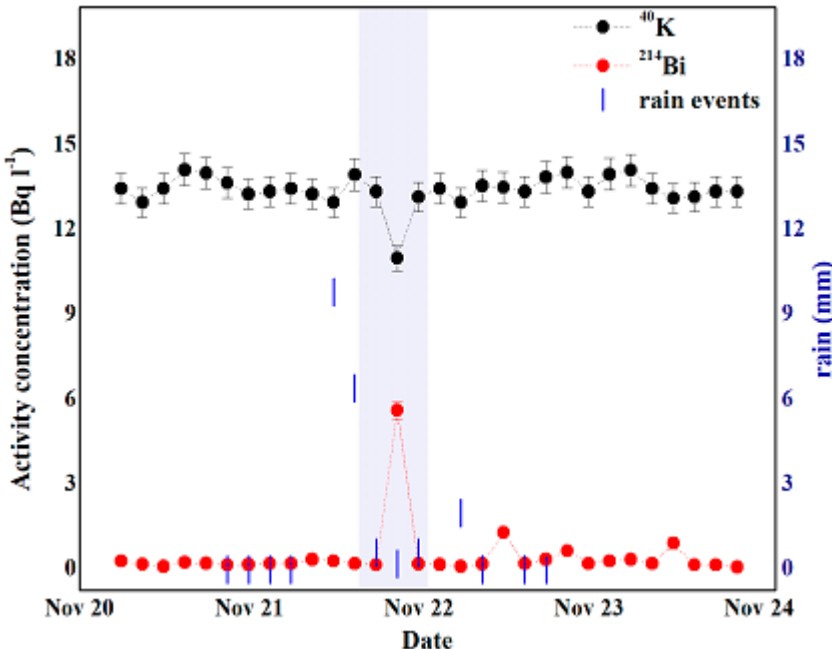

**Figure 6.** The activity concentration of $^{40}$K and $^{214}$Bi for the measuring period (the vertical lines represent the rainfalls).

As concerns the $^{137}$Cs contribution at 662 keV energy, the photopeak was not clearly seen, despite the fact that the proposed method of spectrum analysis provided lower minimum detectable activity (MDA) (for $^{137}$Cs) compared to the work that has been previously done [5]. The lack of $^{137}$Cs contribution in the obtained spectra could be attributed to the high radon concentrations of rainfall and also due to the $^{137}$Cs decay process after 34 years of the Chernobyl accident [39]. The analysis procedure for the reduction of $^{137}$Cs MDA along with the results are given in the next paragraph applying both theoretical and experimental techniques.

## 4. Discussion

### 4.1. MDA of $^{137}$Cs (24 h Acquisition)

The MDA value of the system KATERINA II for $^{137}$Cs was calculated using 24 h measurements during rainfall and a background spectrum (absence of precipitation events) according to Equation (1):

$$\text{MDA}(\text{Bq}/\text{m}^3) = \frac{L_D}{\varepsilon_m \times I_\gamma \times T} \tag{1}$$

where, $L_D$ is the detection limit (in counts), T is the acquisition time (in s), $I_\gamma$ the emission probability of the $\gamma$-ray, and $\varepsilon_m$ is the marine efficiency (in m$^3$). $L_D$ is given as a function of B (net counts) from the equation $L_D = 2.71 + 4.65B^{1/2}$, where B is the net counts at the corresponding gamma-ray energy (foreground with rainfall minus background counts without

rain). The relative uncertainty of the MDA is given applying the typical propagation law of uncertainties and is calculated according to Equation (2):

$$\left(\frac{\delta_{MDA}}{MDA}\right)^2 = \left(\frac{\delta L_D}{L_D}\right)^2 + \left(\frac{\delta \varepsilon_m}{\varepsilon_m}\right)^2 \tag{2}$$

where, the relative uncertainty of $L_D$ is propagated according to Equation (3).

$$\left(\frac{\delta L_D}{L_D}\right) = \left(\frac{2.325\sqrt{B}}{2.7 + 4.65\sqrt{B}} \frac{\sqrt{R + Bg}}{B}\right) \tag{3}$$

where, R and Bg are the measured counts at the gamma-ray energy with and without rainfall, respectively.

Two different time periods (during rainfall) of 12 to 13 December 2019 (total counting rate 19 events per second) and 11 to 12 December 2019 (total counting rate of 15 events per second) were selected to calculate the MDA values, exhibiting the maximum and mean concentrations of radon progenies respectively, along with a background measurement. The spectrum analysis was performed using the SPECTRW software [29], while the MDA calculations were performed according to existing methodologies [28]. The calculation was first performed via peak analysis in the acquired raw data and in the same data after the subtraction of the ambient background radiation, which includes contributions from cosmic radiation, radionuclides present in the air, and the contribution of $^{40}$K, which is always prominent in the seawater. The value of the analyzed total counts drops by a factor of three, (from approximately 20,200 to 6600 counts) after the background subtraction for the measurement at 12 December 2019 and 3.5 times (from 13,300 to 3900) for the measurement at 11/12/2019, respectively.

The background subtraction process leads to a reduction of the detection limit in counts ($L_D$) [28] of 1.75 (for the measurement at 12/12/2019) and 1.85 (for the measurement at 11/12/2019), respectively. This drastic reduction of the $L_D$ value is responsible for the MDA improvement, as the $L_D$ value is directly proportional to the MDA. The statistical uncertainty of the MDA value for $^{137}$Cs in the net spectrum is approximately 5% with highly intense rainfalls and 4% with low-intense rainfall events. The MDA statistical uncertainty value is calculated from the propagation of the statistical uncertainties in the efficiency estimation (around 4%) and the analyzed photopeak counts (using Equation (2)). The statistical uncertainty of the detection limit $L_D$ (in counts) varies from 1–4% in all the calculations (using Equation (3). The final MDA results are given in Table 2.

**Table 2.** MDA values for $^{137}$Cs with and without background subtraction for two different measurements during precipitation events (24 h period with and without background subtraction).

| Measurement Period | MDA for $^{137}$Cs (Bq L$^{-1}$) |
|---|---|
| 12/12/2019 (without background subtraction) | 0.041 |
| 12/12/2019 (background subtraction) | 0.023 |
| 11/12/2019 (without background subtraction) | 0.033 |
| 11/12/2019 (background subtraction) | 0.018 |
| Background | 0.031 |

The MDA values are drastically reduced with the background subtraction rendering this technique a useful tool, allowing low level measurements in the marine environment using the KATERINA II detection system. The MDA results are in agreement with corresponding values of similar detection systems as reported in literature by other groups [40].

### 4.2. Theoretical Estimation of $^{137}$Cs MDA

The MDA of $^{137}$Cs during rainfalls was also theoretically studied applying the FSA technique, using standard spectra for $^{214}$Bi, $^{214}$Pb, and $^{137}$Cs. The reproduced FSA spectrum of the experimental data on a rainfall event during 21 to 22 November 19 (summation of nine acquired spectra normalized to 3 h), is depicted in Figure 7, introducing an activity concentration of 0.3 Bq L$^{-1}$ for the radon progenies ($^{214}$Pb and $^{214}$Bi). The same FSA spectrum is also depicted, introducing an activity concentration of 0.02 Bq L$^{-1}$ for-$^{137}$Cs. The photopeak of $^{137}$Cs becomes apparent in the spectrum for this value (0.02 Bq L$^{-1}$).

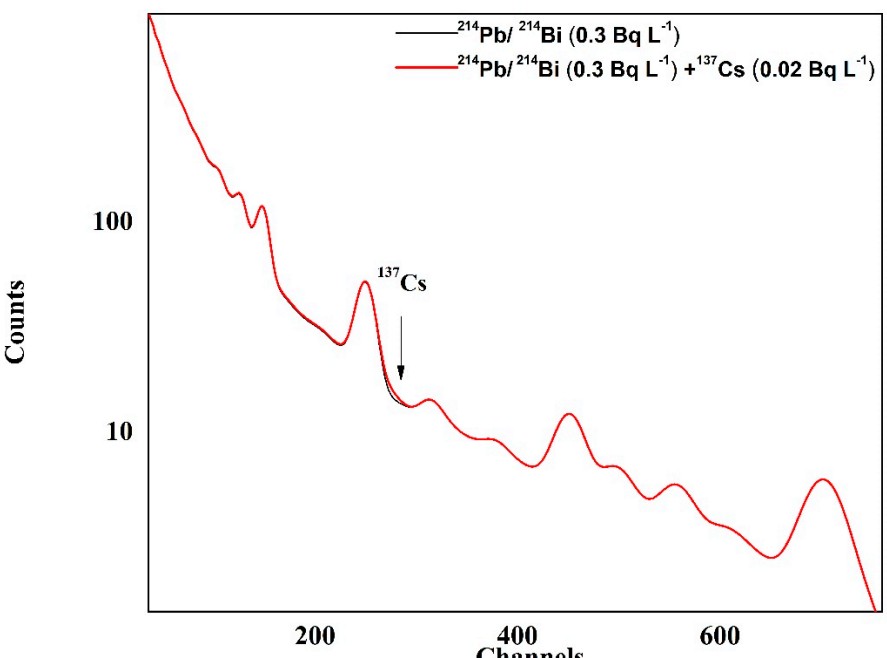

**Figure 7.** The simulated spectrum (at the energy range from threshold to 1000 keV) after the rain on 21 November 2019 (3 h spectrum with activity concentration of radon progenies 0.3 Bq L$^{-1}$) and the $^{137}$Cs contribution of 0.02 Bq L$^{-1}$.

The application of the FSA method, in spectra acquired during rainfall after the background subtraction, provides a methodology for low $^{137}$Cs concentration levels (~0.02 Bq L$^{-1}$) detection, enabling the detection system KATERINA II to quantify low activity concentrations when it operates in the marine environment in incidences of rainfall. Due to the low resolution of the system ($^{214}$Bi contribution in the $^{137}$Cs peak detection area) and the high Compton background from the energetic gamma-rays of radon progenies (such as $^{214}$Bi), the estimated MDA value drastically increases when radon progenies concentration in the marine environment increases (e.g., in case of radon progenies activity concentration above 6 Bq L$^{-1}$, the $^{137}$Cs MDA becomes 0.1 Bq L$^{-1}$).

The obtained MDA values for $^{137}$Cs using both experimental and theoretical techniques are drastically improved, but they are above the expected activity concentration values for $^{137}$Cs (taking into account the decay of $^{137}$Cs), according to previous measurements at the same station in similar meteorological conditions [5].

## 5. Summary and Conclusions

A detailed study was performed applying the KATERINA II detection system to provide quantitative data of various gamma-ray emitters which are present in the seawater before, during and after rainfall. The study of the natural radioactivity variation gives the important result that rainfall events can be detected through radon progenies concentrations, as an increment of $^{214}$Pb and $^{214}$Bi activity concentrations. The detected variation of the activity concentration of radon progenies in the absence of rainfall events, fluctuates

from 0.095 (17%) to 0.53 (15%) Bq L$^{-1}$ for $^{214}$Bi and from 0.14 (30%) to 0.8 (16%) Bq L$^{-1}$ for $^{214}$Pb. The corresponding values in the presence of rainfall events, range from 0.4 (20%) to 5.4 (4%) for $^{214}$Bi and from 0.3 (30%) to 5.3 (4%) Bq L$^{-1}$ for $^{214}$Pb. Therefore, an intense increment of the activity concentration of radon progenies (up to an order of magnitude) was recorded during rainfall.

The integration of the radioactivity sensor to the observation system gave new information for the variation of the level of the γ- activity intensity at North Aegean Sea for the period of rainfalls. During the observation period, the γ- ray intensity was not correlated with the rain intensity (as given from experimental and estimated data). However, the correlation analysis was not performed using experimental data from the same point as with the radioactivity sensor location. The estimated data of the rain intensity were taken from the Poseidon model and the experimental data from the neighbouring stations are provided from the Greek network of the National Observatory of Athens [37]. A more detailed study concerning the correlation of the radionuclide activity concentration variation with environmental parameters from the atmosphere as well as from the sea (like rain intensity, salinity, surface wind etc), is a future application.

The activity concentration of $^{137}$Cs during rainfall in Athos region, is below the MDA of the detection system, and thus, is not detected in the defined time lag of the measurement. However, in marine areas that are strongly affected by nuclear accidents (e.g., Chernobyl, Fukushima), the activity concentration of $^{137}$Cs could be detected in short acquisition periods (less than 3 h). The proposed method could also be applied in the future in areas where previous nuclear accidents took place, in order to detect variations of key radiotracers at neighboring coastal zones, as well as near aquatic ecosystems and waterways that interact with the terrestrial environment (such as streams, submarine groundwater discharge, and estuaries).

It was found that the detected activity concentration of $^{137}$Cs, after intense rainfall events at North Aegean Sea due to the atmospheric wash out process, was reduced by a factor of 3 compared to measurements in 2000 [5], where a similar detection system was integrated in the same buoy station of the POSEIDON network. Furthermore, the activity concentrations of radon progenies were found in radioactive equilibrium in most cases after such rainfall events, exhibiting high concentration values (especially if the origin and direction of the respective cloud were from a terrestrial environment). The intense rainfall events enrich the seawater mass with high concentrations of radon progenies (as tracer of rainwater mass) and lower the values of $^{40}$K (as tracer of the seawater) compared with the typical values of $^{40}$K in the sea. Thus, the dilution process of rainwater in the sea surface layer should also be studied in the future, combining the aforementioned tracers (radon progenies and $^{40}$K) which are given directly quantitatively by the KATERINA II underwater detection system.

**Author Contributions:** Data curation, A.I.; Formal analysis, C.T. and E.G.A.; Funding acquisition, A.I.; Investigation, C.T., E.G.A., D.B., S.A. and L.P.; Methodology, C.T., E.G.A., D.B. and S.A.; Resources, A.I.; Software, E.G.A. and S.A.; Supervision, C.T.; Validation, L.P. and A.I.; Visualization, A.I.; Writing—original draft, C.T.; Writing—review & editing, C.T., E.G.A. and L.P. All authors have read and agreed to the published version of the manuscript.

**Funding:** This work was supported by the MARRE project through National Strategic Reference Framework (NSRF) 2014–2020 co-financed by Greece and the European Union (European Social Fund ESF).

**Institutional Review Board Statement:** The study was conducted according to the MARRE project and approved for submission by the Institutional Director of the Institute of Oceanography (1/9/2020).

**Informed Consent Statement:** Informed consent was obtained from all subjects involved in the study.

**Data Availability Statement:** The data are available on request. The data presented in this study are available on request from the corresponding author. The data are not publicly available since MARRE is an on-going project.

**Acknowledgments:** This work was supported by the MARRE project through National Strategic Reference Framework (NSRF) 2014–2020 co-financed by Greece and the European Union (European Social Fund ESF). The authors would like to acknowledge the POSEIDON group and especially Nikolaos Zisis for the continuous support and for the integration exercises of the sensor to the monitoring network system. The authors would like to thank the divers Vassilis Stasinos and Vassilis Mpampas for the short term cruise to access the observing station at Athos. The authors would also want to acknowledge the crew of the research vessels AEGEAO for installation exercises as well as for the system maintenance during the cruise in the frame of MARRE project (February 2020). The authors would like also to acknowledge Spiridon Velanas for designing the graph of the observing system. At last, the authors would like to thank the group at National Observation of Athens for providing meteorological data of the neighboring stations for the rainfall events during the monitoring period.

**Conflicts of Interest:** There is no conflict of interest.

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
