# Peer review of "Radioactivity Monitoring at North Aegean Sea Integrating In-Situ Sensor in an Ocean Observing Platform"

_jmse, doi:10.3390/jmse9010077_

Round 1
Reviewer 1 Report
This manuscript is not answered to my prevous comments. Therefore, I cannot permit publication to the journal.
Author Response
First Round
Reviewer 1:
The authors should clearly mention the aim of this paper.
1. Identifying the event of rainfall? Improving the quantatitativeness of Cs-137?
New text (page 1 lines 31-33) is introduced and replaced in the introduction paragraph.
The sentence “Radioactivity monitoring tools and methods are tested in many marine areas of the world the last years [1-6] depending on the selected area, topography and contamination level” is replaced with:
"Different in situ methodologies and tools regarding radioactivity monitoring are tested in many marine areas of the world the last years [1-11], depending on the selected area, topography and contamination level”.
The following text (page2 lines 57-62) is added to better define the main goal of this work:
The main goal of this work was to study the atmospheric wash out due to rainfalls, regarding natural (e.g. radon progenies) and artificial radioactivity (e.g. 137Cs), as well as to improve the minimum detectable activity (MDA) of the detection system in order to detect low concentrations of the artificial 137Cs. For this purpose, a new methodology is developed which relies on spectra subtraction to minimize the measurement background prior the analysis and utilization of the Full Spectrum Analysis (FSA) technique for the 137Cs concentration values estimation.
2. Originality and improvement compared to previous studies are unclear. Why the previous methods are insufficient?
The following text (page2 lines 47-56) is added:
Data regarding the utilization of radiotracers for meteorological studies (e.g. correlation rainfall events with wind speed direction) is scarce [6,12,13], especially in the marine environment, although it provides ideal conditions for such studies, eliminating factors adding complexity (e.g. wind, temperature variations). In these studies, strong correlation of rainfall events with radon progenies activity concentrations have been observed [6,13] depending on the passage and origin of the rainfall (if it comes from the terrestrial or marine environment). In this study the correlation analysis between radioactivity data and rainfall parameters of neighboring stations was performed. Moreover, the phenomenon of surface salinity variations induced by rainfall dilution [14,15] was observed via radioactivity data using as tracer 40K.
3. What were the noteworthy points of the data in the present study? Additionally, how those data can be used for understanding the dynamics of radionuclides and/or other materials?
The following text (page14 lines 402-412) is added to the summary section:
It was found that the activity concentration of 137Cs, produced after intense rainfall events at North Aegean Sea due to the atmospheric wash out process, was reduced by a factor of 4 compared with measurements in 2000 where similar detection system was integrated in the same buoy station of the POSEIDON network [5]. Furthermore, the activity concentration of radon progenies were in equilibrium in most cases after such rainfall events exhibiting high concentration values (especially, if the origin and direction of the respective cloud was from a terrestrial environment). The intense rainfall events enrich the seawater mass with high values of radon progenies (as tracer of rainwater mass) and lower values of 40K (as tracer of the seawater) compared with typical values of the sea. Thus, the dilution process of rainwater in the sea surface layer should be also studied in the future combing the aforementioned tracers (radon progenies and 40K) as given directly in a quantitative manner by the KATERINA II detection underwater system.
4. The authors should separate the sentences related to 'method' from the result section.
The following sentence (page 4 lines 138-139) was moved from the results to the “method” section:
The first step in the analysis is the energy calibration as given from existing methodology at the laboratory [7,12].
Other text was moved from the Results Section to Methods Section (please see revised manuscript with “track changes”).
5. There was few explanations related to statistical analyses.
The following text (page 4 lines 146-156) is added/modified in the “Methods” section:
The activity concentration of radon progenies (214Bi, 214Pb) was calculated in rainfall events, using experimental and theoretical techniques. The SPECTRW software [22] was utilized in the spectra analysis. The activity concentration of radon progenies was deduced taking into account the detection efficiency of the system, the detected counting rate, the acquisition time and the emission probability of the detected gamma-rays [21,31]. The statistical uncertainty of the simulation runs for the detection efficiency of the system was kept below 3% in the energy regions of the photopeaks, in all cases. Thus, the total uncertainty budget is calculated using the typical propagation law of relative uncertainties taking into account the uncertainties of the system efficiency (3%) and the statistical counting rate uncertainty in the analyzed photopeaks (4-39%). The analysis is performed assuming homogenized enrichment of radon progenies in the seawater and rapid dilution process in the time lag window of the detection system (3h).
The following text (page 11-12, lines 329-342) is added/modified in the “Discussion” section:
The value of the analyzed total counts drops 3 times from approximately 20200 to 6600 counts after the background subtraction for the measurement at 12/12 and 3.5 times from 13300 to 3900 for the measurement at 11/12, respectively, leading to a reduction in the detection limit in counts (LD) [16] of 1.75 (for the measurement at 12/12) and 1.85 (for the measurement at 11/12) respectively. This drastic reduction of the LD value is responsible for the MDA improvement, as the LD value is directly proportional to the MDA. Although the gain in the MDA is high, it is followed by an increase in the total statistical uncertainty of the MDA calculation, due to the increase to the statistical uncertainty arising from the background subtraction. The statistical uncertainty of the MDA values rises to 5% (for the measurement at 11/12) and 6% (for the measurement at 12/12) respectively, while the statistical uncertainty of the MDA value for 137Cs in the background spectrum is approximately 3%. The MDA statistical uncertainty value is calculated from the propagation of the statistical uncertainties in the efficiency estimation (around 3%) and the analyzed counts. The detection limit (in counts) statistical uncertainty is around 1% in all the calculations without the background subtraction.
6. The information related to water quality such as salinity was scarcely discussed.
The following text (page 10 line 294-295) is added:
The 40K data exhibit stable activity concentration values during dry periods representing the salinity values of the seawater (mean value of 37 ‰ according to buoy salinity data).
---------------------------------------------------
Round 2
Reviewer 1
a. The authors mentioned that one of the goals is to identify the natural radioactivity. In this case, they must explain importance of the natural radioactivity with some examples in Introduction section. They only explained about the importance of artificial radioactivity, such as radioactivity derived from nuclear accidents. Additionally, for another goal, identifying background levels of Cs-137, the authors had better explain its importance in the Introduction section.
The following information was updated in the introduction section:
Assuming that the background gamma-ray radiation level (which depends mostly on the water salinity) is constant at a specific area at the open sea, the fluctuations of the gross counting rate monitored in seawater are mainly caused by rainfall due to the atmospheric wash out of the various natural radionuclides (in the absence of a radiological incidence). Radon as a natural radioactive inert gas and non-chemically reactive element is studied extensively in the terrestrial and the marine environment concerning NORM and TENORM studies [12]. One of its isotopes, 222Rn, although it is not a gamma-emitter, it can be detected via gamma-ray spectrometry from its progenies (214Bi and 214Pb). These two radionuclides have been extensively utilized as radiotracers in oceanography applications [13,14], such as water masses mixing, radioprotection purposes in non-nuclear industries (fertilizers, oil exploration, oil and construction industries), pockmarks [15], mud volcanoes [16], and submarine groundwater discharge identification [17,18]. Moreover, the monitoring of natural and artificial radioactivity variations during rainfall at the sea can provide important information in critical hazards (e.g. fires, floods). The radionuclides 214Bi and 214Pb have been used as radiotracers also in different scientific fields.
The following references are added in the reference list (line 506 and line 521, respectively).
Povinec, P.P., Eriksson, M., Scholten, J., Betti, M., Marine Radioactivity Analysis, In Handbook of Radioactivity Analysis (Third Edition), Chapter 12, edited by Michael F. L'Annunziata,, Academic Press, Amsterdam, 2012, p. 769-832.
Tsabaris, C., Scholten, J., Karageorgis, A. P., Comanducci, J.-F., Georgopoulos, D., Liong Wee Kwong, L., Patiris, D. L., Papathanassiou, E., Underwater in situ measurements of radionuclides in selected submarine groundwater springs, Mediterranean Sea, Radiat. Prot. Dosim. 2010, 142, 2-4, 273-28.
b. The explanation of water quality is too rough. A concentration of potassium in seawater is influenced by salinity, resulting that K-40 activity is probably influenced by the salinity. Therefore, the relationship between the salinity and K-40 activity should be analyzed and described more carefully when the authors try to discuss the dynamics of K-40.
The following information was updated in the introduction section:
However, the activity concentration of 40K is drastically decreased during and after strong precipitation due to the rapid mixing processes of the rainwater with the seawater. A typical example is given in Fig. 6, where the influence of precipitation is clearly observed from the KATERINA II detection system by the decrease of the measured 40K activity concentration (almost 30% compared with the dry periods) during the precipitation event at 22nd of November 2019. The same behavior was observed during the rainfall event at 12th of December. This precipitation had an amount of 9.2 mm and the rainwater was enriched with radon progenies (214Bi) concentration of (5.4 ± 0.3) Bq l-1. The calculated salinity values using the 40K activity concentrations ranged from 30 to 39 psu, while similar values ranging from 33 to 37 psu were also obtained from the buoy CTD measurements (in 1m depth) in the same station. This was expected as the 40K concentration is very well correlated with salinity values in the seawater [13]. The salinity data (both the values and the observed variations) agree with salinity values obtained in the past in the same station [38]. The decrease of the 40K activity concentration could be attributed to the dilution of seawater through mixing with different water masses either due to the large amount of rainwater or a passage of a seawater current mass (e.g. Black Sea Water) which has been observed in the past [38].
c. I would like to recommend the check of English-writing.
English are checked and revised by an anonymous colleague.

Reviewer 2 Report
no comments
Author Response
Reviewer 2
This is excellent work in a long continuing research project spanning two decades. What impresses me the most is detection of the naturally occurring radioactive isotopes before and after rainfall. As well, the lowering of the detection limit of 137Cs is quite notable. It would be nice to see if this could be applied to waterways or the ocean near Fukushima or even in water systems in parts of the Ukraine sometime in the future.
The following text (page 14 lines 396-400) is added at the conclusion-summary paragraph:
The proposed method could be applied in the future in areas where previous nuclear accidents took place in order to detect the variations of 137Cs activity concentration at the neighboring coastal zone as well as near aquatic ecosystems and waterways that interact with the terrestrial environment (such as streams, submarine groundwater discharges, estuaries).
Reviewer 3 Report
- Regarding MDA calculation with the background subtraction method.
Could you please check the "standard deviation" of background counts for "the background subtraction method"?
The standard deviation should be derived through error propagation when background counts are subtracted. How did you determine the standard deviation for LD calculation?
Author Response
Reviewer 3
- a) To apply the correction factor to assess the radioactivity of radon progenies such as 214Bi and 214Pb that have been into seawater by the atmospheric wash out, these radon progenies must be homogeneously mixed in the seawater around the detector. Could you please mention additional comments about this for the readers? If this explanation is not available, it would be also good to mention the assessment conditions as the assumption that radon progenies are homogeneously distributed around the detector.
The following assumption is given to the revised manuscript (page 5 lines 162):
The typical analysis technique is the photopeak analysis.
The following assumption is given to the revised manuscript (page 6 lines 178-181):
Nevertheless, it is important to mention that in the simulations the radionuclides are considered to be homogeneously distributed in a large (with radius of 50 cm to 200 cm depending on the detected gamma-ray energy) spherical volume surrounding the detector [28,29].
- b) Regarding MDA calculation with the background subtraction method.
How did you estimate the standard deviation of background counts for calculating the MDAs with ‘background subtraction’ method? Could you please describe in the manuscript which factors made the differences between the MDAs with and without ‘background subtraction’ method?
The following text (page 11 lines 328-347) is added in the revised manuscript:
The following text is added:
The MDA for 137Cs of the system KATERINA II was calculated for measured spectra of 24 hours during rainfall and a background spectrum (absence of precipitation events) according to the Eq.1:
(1)
where, LD is the detection limit (in counts), T is the acquisition time (in s), Iγ the emission probability of the γ-ray and εm is the marine efficiency (in m3). LD is given as a function of B (net counts) from the equation LD= 2.71+4.65B1/2, where B is the net counts at the corresponding gamma-ray energy (foreground with rain minus background counts without rain). The relative uncertainty of the MDA is given applying the typical propagation law of uncertainties and is calculated according to Eq. 2:
(2)
where, the relative uncertainty of LD is propagated according to the Eq. 3
(3)
where R and Bg are the measured counts at the gamma-ray energy with and without rainfall, respectively.
At the end of section 4.1 the following text is replaced:
The background subtraction process leads to a reduction of the detection limit in counts (LD) [23] of 1.75 (for the measurement at 12/12/19) and 1.85 (for the measurement at 11/12/19), respectively. This drastic reduction of the LD value is responsible for the MDA improvement, as the LD value is directly proportional to the MDA. The statistical uncertainty of the MDA value for 137Cs in the net spectrum is approximately 4% with high-intense rainfalls and 4% with low-intense rainfall events. The MDA statistical uncertainty value is calculated from the propagation of the statistical uncertainties in the efficiency estimation (around 4%) and the analyzed photopeak counts (using Eq.2). The statistical uncertainty of the detection limit LD (in counts) varies from 1-4% in all the calculations (using Eq. 3). The final MDA results are given in Table 2.
The MDA values are drastically reduced with the background subtraction rendering this technique a useful tool, allowing low level measurements in the marine environment using the KATERINA II detection system. The MDA results are in agreement with corresponding values of similar detection systems as reported in literature by other groups [40].

Round 2
Reviewer 1 Report
I have no comments.
This manuscript is a resubmission of an earlier submission. The following is a list of the peer review reports and author responses from that submission.
Round 1
Reviewer 1 Report
The manuscript entitled 'Radioactivity monitoring at North Aegean Sea integrating in-situ sensor in an ocean observing platform' shows the result of remote sensor measuremnt of radioactivity in seawater in the specific site. Although this manuscript may include a novel technique for the monitoring encvironmental radioactivity, there are many concerns related to originality and adequacy of data and discussion. Therefore, I strongly would like to remake the manuscript and data analysis.
The main comments are described as follows:
- The authors should clearly mention the aim of this paper. Identifing the event of rainfall? Improving the quantatitativeness of Cs-137?
- Originality and improvement compared to previous studies are unclear. Why the previous methods are insuficient?
- What were the noteworthy points of the data in the present study? Addtionally, how those data can be used for understanding the dynamics of radionuclides and/or other materials?
- The auhors should separate the sentences related to 'method' from the result section.
- There was few explanations related to statistical analyses.
- The infomation related to water quality such as salinity was scarecely discussed.
Author Response
2 December 2020
Dear Editor,
Please find attached the revised version regarding the paper entitled:
“Radioactivity monitoring at North Aegean Sea integrating in-situ sensor in an ocean observing platform”
authored by
Chirstos Tsabaris , Efrossyni G. Androulakaki, Dionisios Ballas, Stylianos Alexakis, Leonidas Perivoliotis, Athanasia Iona,
which is submitted for consideration and possible publication in the journal “Journal of Marine Science and Engineering”.
The revisions are made according to the reviewer comments and their instructions. The text is revised with “Track Changes”, while changes are explained in the attached files.
All authors would like to thank reviewers for the valuable comments they made to improve the manuscript.
Looking forward to hearing from you, and being at your disposal for any additional information or material concerning the submitted paper, I remain
Sincerely yours,
Christos Tsabaris
Corresponding Author
Reviewer 1:
The authors should clearly mention the aim of this paper.
- Identifying the event of rainfall? Improving the quantatitativeness of Cs-137?
New text (page 1 lines 31-33) is introduced and replaced in the introduction paragraph.
The sentence “Radioactivity monitoring tools and methods are tested in many marine areas of the world the last years [1-6] depending on the selected area, topography and contamination level” is replaced with:
"Different in situ methodologies and tools regarding radioactivity monitoring are tested in many marine areas of the world the last years [1-11], depending on the selected area, topography and contamination level”.
The following text (page2 lines 57-62) is added to better define the main goal of this work:
The main goal of this work was to study the atmospheric wash out due to rainfalls, regarding natural (e.g. radon progenies) and artificial radioactivity (e.g. 137Cs), as well as to improve the minimum detectable activity (MDA) of the detection system in order to detect low concentrations of the artificial 137Cs. For this purpose, a new methodology is developed which relies on spectra subtraction to minimize the measurement background prior the analysis and utilization of the Full Spectrum Analysis (FSA) technique for the 137Cs concentration values estimation.
- Originality and improvement compared to previous studies are unclear. Why the previous methods are insufficient?
The following text (page2 lines 47-56) is added:
Data regarding the utilization of radiotracers for meteorological studies (e.g. correlation rainfall events with wind speed direction) is scarce [6,12,13], especially in the marine environment, although it provides ideal conditions for such studies, eliminating factors adding complexity (e.g. wind, temperature variations). In these studies, strong correlation of rainfall events with radon progenies activity concentrations have been observed [6,13] depending on the passage and origin of the rainfall (if it comes from the terrestrial or marine environment). In this study the correlation analysis between radioactivity data and rainfall parameters of neighboring stations was performed. Moreover, the phenomenon of surface salinity variations induced by rainfall dilution [14,15] was observed via radioactivity data using as tracer 40K.
- What were the noteworthy points of the data in the present study? Additionally, how those data can be used for understanding the dynamics of radionuclides and/or other materials?
The following text (page14 lines 402-412) is added to the summary section:
It was found that the activity concentration of 137Cs, produced after intense rainfall events at North Aegean Sea due to the atmospheric wash out process, was reduced by a factor of 4 compared with measurements in 2000 where similar detection system was integrated in the same buoy station of the POSEIDON network [5]. Furthermore, the activity concentration of radon progenies were in equilibrium in most cases after such rainfall events exhibiting high concentration values (especially, if the origin and direction of the respective cloud was from a terrestrial environment). The intense rainfall events enrich the seawater mass with high values of radon progenies (as tracer of rainwater mass) and lower values of 40K (as tracer of the seawater) compared with typical values of the sea. Thus, the dilution process of rainwater in the sea surface layer should be also studied in the future combing the aforementioned tracers (radon progenies and 40K) as given directly in a quantitative manner by the KATERINA II detection underwater system.
- The authors should separate the sentences related to 'method' from the result section.
The following sentence (page 4 lines 138-139) was moved from the results to the “method” section:
The first step in the analysis is the energy calibration as given from existing methodology at the laboratory [7,12].
Other text was moved from the Results Section to Methods Section (please see revised manuscript with “track changes”).
- There was few explanations related to statistical analyses.
The following text (page 4 lines 146-156) is added/modified in the “Methods” section:
The activity concentration of radon progenies (214Bi, 214Pb) was calculated in rainfall events, using experimental and theoretical techniques. The SPECTRW software [22] was utilized in the spectra analysis. The activity concentration of radon progenies was deduced taking into account the detection efficiency of the system, the detected counting rate, the acquisition time and the emission probability of the detected gamma-rays [21,31]. The statistical uncertainty of the simulation runs for the detection efficiency of the system was kept below 3% in the energy regions of the photopeaks, in all cases. Thus, the total uncertainty budget is calculated using the typical propagation law of relative uncertainties taking into account the uncertainties of the system efficiency (3%) and the statistical counting rate uncertainty in the analyzed photopeaks (4-39%). The analysis is performed assuming homogenized enrichment of radon progenies in the seawater and rapid dilution process in the time lag window of the detection system (3h).
The following text (page 11-12, lines 329-342) is added/modified in the “Discussion” section:
The value of the analyzed total counts drops 3 times from approximately 20200 to 6600 counts after the background subtraction for the measurement at 12/12 and 3.5 times from 13300 to 3900 for the measurement at 11/12, respectively, leading to a reduction in the detection limit in counts (LD) [16] of 1.75 (for the measurement at 12/12) and 1.85 (for the measurement at 11/12) respectively. This drastic reduction of the LD value is responsible for the MDA improvement, as the LD value is directly proportional to the MDA. Although the gain in the MDA is high, it is followed by an increase in the total statistical uncertainty of the MDA calculation, due to the increase to the statistical uncertainty arising from the background subtraction. The statistical uncertainty of the MDA values rises to 5% (for the measurement at 11/12) and 6% (for the measurement at 12/12) respectively, while the statistical uncertainty of the MDA value for 137Cs in the background spectrum is approximately 3%. The MDA statistical uncertainty value is calculated from the propagation of the statistical uncertainties in the efficiency estimation (around 3%) and the analyzed counts. The detection limit (in counts) statistical uncertainty is around 1% in all the calculations without the background subtraction.
- The information related to water quality such as salinity was scarcely discussed.
The following text (page 10 line 294-295) is added:
The 40K data exhibit stable activity concentration values during dry periods representing the salinity values of the seawater (mean value of 37 ‰ according to buoy salinity data).
Reviewer 2
This is excellent work in a long continuing research project spanning two decades. What impresses me the most is detection of the naturally occurring radioactive isotopes before and after rainfall. As well, the lowering of the detection limit of 137Cs is quite notable. It would be nice to see if this could be applied to waterways or the ocean near Fukushima or even in water systems in parts of the Ukraine sometime in the future.
The following text (page 14 lines 396-400) is added at the conclusion-summary paragraph:
The proposed method could be applied in the future in areas where previous nuclear accidents took place in order to detect the variations of 137Cs activity concentration at the neighboring coastal zone as well as near aquatic ecosystems and waterways that interact with the terrestrial environment (such as streams, submarine groundwater discharges, estuaries).
Reviewer 3
- a) To apply the correction factor to assess the radioactivity of radon progenies such as 214Bi and 214Pb that have been into seawater by the atmospheric wash out, these radon progenies must be homogeneously mixed in the seawater around the detector. Could you please mention additional comments about this for the readers? If this explanation is not available, it would be also good to mention the assessment conditions as the assumption that radon progenies are homogeneously distributed around the detector.
The following assumption is given to the revised manuscript (page 5 lines 162):
The typical analysis technique is the photopeak analysis.
The following assumption is given to the revised manuscript (page 6 lines 178-181):
Nevertheless, it is important to mention that in the simulations the radionuclides are considered to be homogeneously distributed in a large (with radius of 50 cm to 200 cm depending on the detected gamma-ray energy) spherical volume surrounding the detector [28,29].
- b) Regarding MDA calculation with the background subtraction method.
How did you estimate the standard deviation of background counts for calculating the MDAs with ‘background subtraction’ method? Could you please describe in the manuscript which factors made the differences between the MDAs with and without ‘background subtraction’ method?
The following text (page 11 lines 328-347) is added in the revised manuscript:
The value of the analyzed total counts drops 3 times from approximately 20200 to 6600 counts after the background subtraction for the measurement at 12/12 and 3.5 times from 13300 to 3900 for the measurement at 11/12, respectively, leading to a reduction in the detection limit in counts (LD) [16] of 1.75 (for the measurement at 12/12) and 1.85 (for the measurement at 11/12) respectively. This drastic reduction of the LD value is responsible for the MDA improvement, as the LD value is directly proportional to the MDA. Although the gain in the MDA is high, it is followed by an increase in the total statistical uncertainty of the MDA calculation, due to the increase to the statistical uncertainty arising from the background subtraction. The statistical uncertainty of the MDA values rises to 5% (for the measurement at 11/12) and 6% (for the measurement at 12/12) respectively, while the statistical uncertainty of the MDA value for 137Cs in the background spectrum is approximately 3%. The MDA statistical uncertainty value is calculated from the propagation of the statistical uncertainties in the efficiency estimation (around 3%) and the analyzed counts. The detection limit (in counts) statistical uncertainty is around 1% in all the calculations without the background subtraction. The final MDA results are given in Table 2.
The MDA values are drastically reduced with the background radiation subtraction rendering this technique a useful tool, allowing low level measurements in the marine environment using the KATERINA II detection system. The MDA results are in agreement with corresponding values of similar detection systems as reported in literature by other groups [32].

Reviewer 2 Report
This is excellent work in a long continuing research project spanning two decades. What impresses me the most is detection of the naturally occurring radioactive isotopes before and after a rainfall. As well, the lowering of the detection limit of 137Cs is quite notable. It would be nice to see if this could be applied to waterways or the ocean near Fukushima or even in water systems in parts of the the Ukraine sometime in the future. All in all, very well presented.
Author Response

(The authors gave the same response as above.)

Reviewer 3 Report
It seems that this paper can be used as a valuable reference for understanding the in-situ gamma-ray energy spectrum in seawater. In particular, it is interesting to evaluate changes in natural radioactivity in the sea due to wash-out with wind direction. I would like to just present two issues as follow.
- To apply the correction factor to assess the radioactivity of radon progenies such as 214Bi and 214Pb that have been into seawater by the atmospheric wash out, these radon progenies must be homogeneously mixed in the seawater around the detector. Could you please mention additional comments about this for the readers? If this explanation is not available, it would be also good to mention the assessment conditions as the assumption that radon progenies are homogeneously distributed around the detector.
- Regarding MDA calculation with the background subtraction method.
How did you estimate the standard deviation of background counts for calculating the MDAs with ‘background subtraction’ method? Could you please describe in the manuscript which factors made the differences between the MDAs with and without ‘background subtraction’ method?
Author Response
Reviewer 3
- a) To apply the correction factor to assess the radioactivity of radon progenies such as 214Bi and 214Pb that have been into seawater by the atmospheric wash out, these radon progenies must be homogeneously mixed in the seawater around the detector. Could you please mention additional comments about this for the readers? If this explanation is not available, it would be also good to mention the assessment conditions as the assumption that radon progenies are homogeneously distributed around the detector.
The following assumption is given to the revised manuscript (page 5 lines 162):
The typical analysis technique is the photopeak analysis.
The following assumption is given to the revised manuscript (page 6 lines 178-181):
Nevertheless, it is important to mention that in the simulations the radionuclides are considered to be homogeneously distributed in a large (with radius of 50 cm to 200 cm depending on the detected gamma-ray energy) spherical volume surrounding the detector [28,29].
- b) Regarding MDA calculation with the background subtraction method.
How did you estimate the standard deviation of background counts for calculating the MDAs with ‘background subtraction’ method? Could you please describe in the manuscript which factors made the differences between the MDAs with and without ‘background subtraction’ method?
The following text (page 11 lines 328-347) is added in the revised manuscript:
The value of the analyzed total counts drops 3 times from approximately 20200 to 6600 counts after the background subtraction for the measurement at 12/12 and 3.5 times from 13300 to 3900 for the measurement at 11/12, respectively, leading to a reduction in the detection limit in counts (LD) [16] of 1.75 (for the measurement at 12/12) and 1.85 (for the measurement at 11/12) respectively. This drastic reduction of the LD value is responsible for the MDA improvement, as the LD value is directly proportional to the MDA. Although the gain in the MDA is high, it is followed by an increase in the total statistical uncertainty of the MDA calculation, due to the increase to the statistical uncertainty arising from the background subtraction. The statistical uncertainty of the MDA values rises to 5% (for the measurement at 11/12) and 6% (for the measurement at 12/12) respectively, while the statistical uncertainty of the MDA value for 137Cs in the background spectrum is approximately 3%. The MDA statistical uncertainty value is calculated from the propagation of the statistical uncertainties in the efficiency estimation (around 3%) and the analyzed counts. The detection limit (in counts) statistical uncertainty is around 1% in all the calculations without the background subtraction. The final MDA results are given in Table 2.
The MDA values are drastically reduced with the background radiation subtraction rendering this technique a useful tool, allowing low level measurements in the marine environment using the KATERINA II detection system. The MDA results are in agreement with corresponding values of similar detection systems as reported in literature by other groups [32].
Round 2
Reviewer 1 Report
Thank you for replying to my comments. The authors revised hardly, but there are still insufficeint parts in their manuscript.
- The authors mentionted that one of the goals is to identify the natural radioactivity. In tis case, they must explain importances of the natural radioactivity with some examples in Introduction section. They only explained about the impotance of artificial radioactivity, such as radioactivity derivaed from nuclear accidents. Addtionally, for the another goal, identyfing background levels of Cs-137, the authors had better explain its importance in the Introduction section.
- The explanation of water quality is too rough. A concentration of potasium in seawater is influenced by salinity, resulting that K-40 activity is probably influenced by the salinity. Thererfore, the relationship between the salinity and K-40 activity should be analyzed and described more carefully when the authors try to discuss the dynamics of K-40.
- I would like to recomend the check of English-writing.